# CE-BART: Cause-and-Effect BART for Visual Commonsense Generation

**DOI:** 10.3390/s22239399

**Published:** 2022-12-02

**Authors:** Junyeong Kim, Ji Woo Hong, Sunjae Yoon, Chang D. Yoo

**Affiliations:** 1Department of AI, Chung-Ang University, Seoul 06974, Republic of Korea; 2School of Electrical Engineering, Korea Advanced Institute of Science and Technology, Daejeon 34141, Republic of Korea

**Keywords:** deep learning, visual–language reasoning, visual commonsense generation, video-grounded dialogue, VisualCOMET, AVSD

## Abstract

“A Picture is worth a thousand words”. Given an image, humans are able to deduce various cause-and-effect captions of past, current, and future events beyond the image. The task of visual commonsense generation has the aim of generating three cause-and-effect captions for a given image: (1) what needed to happen before, (2) what is the current intent, and (3) what will happen after. However, this task is challenging for machines, owing to two limitations: existing approaches (1) directly utilize conventional vision–language transformers to learn relationships between input modalities and (2) ignore relations among target cause-and-effect captions, but consider each caption independently. Herein, we propose Cause-and-Effect BART (CE-BART), which is based on (1) a structured graph reasoner that captures intra- and inter-modality relationships among visual and textual representations and (2) a cause-and-effect generator that generates cause-and-effect captions by considering the causal relations among inferences. We demonstrate the validity of CE-BART on the VisualCOMET and AVSD benchmarks. CE-BART achieved SOTA performance on both benchmarks, while an extensive ablation study and qualitative analysis demonstrated the performance gain and improved interpretability.

## 1. Introduction

Visual Commonsense Generation (VCG) [1] is a challenging task that requires the generation of commonsense and cause-and-effect captions regarding visual and textual information. To be specific, given a still image and a description of the event shown in that image, the goal is to understand the cause-and-effect relations within the event and generate free-form natural language sentences that describe the inferred past/future events and the present intents of characters in the image. For example, in Figure 1, given the image on the left of a woman approaching a man at a table, the agent generates three kinds of cause-and-effect captions: (1) sometime in the past, she walked into the room and saw a man sitting at the table; (2) the intent of the woman is to talk to the man; (3) sometime in the future, she will sit down at the table and speak with him about a serious topic. While reasoning about the rich dynamic story of the visual scene is easy for humans, it is difficult for machines since it requires a higher-order cognitive-level understanding of the world.

In recent years, several visual reasoning tasks [2,3,4] were proposed and drew attention in the computer vision and natural language processing communities. To elaborate on a few such proposals, the Visual-Question-Answering (VQA) task defines a question-answering paradigm as a test to measure a machine’s reasoning abilities for a given image or video. The Visual Dialog (VisDial) task asks a series of questions in the form of dialogue grounded in an image or video. The Visual-Commonsense-Reasoning (VCR) task further requires the machine to provide a rationale explaining why its answer is correct. While the above visual reasoning tasks are defined as recognition-level understanding and only consider the concepts and relations *within* the provided image or video, VCG focuses on reasoning about the rich cognitive-level dynamic story that goes beyond the directly visible contents by requiring cause-and-effect caption generation. Piaget’s cognitive development theory [5] describes the drive of human intelligence to know in two forms, that is as states and transformations, suggesting that people must possess functions to represent both static and transformational aspects of reality. If the former tasks represent reasoning in a stationary situation, then VCG represents reasoning in a transforming situation. Hence, the research on VCG opens the door for a major leap from recognition-level understanding to cognition-level reasoning.

Only a few works on VCG have been published. Park et al. [1] constructed a benchmark for visual commonsense generation, VisualCOMET, and proposed the baseline method. Xing et al. [6] proposed Knowledge-Enhanced BART (KM-BART), which leverages knowledge from external corpora to pre-train the BART. Previous approaches only operated on the conventional learning scheme of visual and textual information, overlooking the distinctiveness of cause-and-effect generation. Two major limitations of previous approaches are that they (1) directly utilized conventional vision–language transformers to learn relationships between input modalities and (2) ignored relations among target captions, but considered each caption independently. Due to the first limitation, previous approaches ignored the intra- and inter-modality relationships, which have proven to be beneficial to transformer-based generation [7]. Due to the second limitation, the existing models pay no attention to the intrinsic structure of the task or dataset. As the goal of VCG is to generate cause-and-effect captions, it is essential to consider causal relations among each inference for *before, intent*, and *after*. While existing approaches consider these inferences as separate cases and are trained independently, we argue that the generation of these three captions should be considered holistically.

In this paper, we address the aforementioned limitations with our novel Cause-and-Effect BART (CE-BART), which is composed of (1) a Structured Graph Reasoner (SGR) and (2) a Cause-and-Effect Generator (CEG). The SGR first builds semantic graphs for each modality to interpret the intra-modality relationships from the spatial or token domain via graph structures, then it captures higher-order semantic relations among graphs (i.e., inter-modality relationships) via tripartite graph attention and strengthens the multimodal graph representations. As SGR comprehends the intra- and inter-relationships interspersed in multimodal representations beforehand, the latter workload of the transformer-based CEG is unburdened, allowing it to focus more on understanding the commonsense and cause-and-effect inference of the given input. The CEG generates cause-and-effect captions for *before, intent*, and *after* situations. While all existing approaches for visual commonsense generation are trained to generate each cause-and-effect caption separately (i.e., there are no connections between the generation of *before, intent*, and *after*, even for the same image), the proposed CEG infers all three cause-and-effect captions holistically by considering the causal relations. It consists of one transformer encoder for modeling multimodal representations and three transformer decoders each for generating the *before, intent*, and *after* captions. To consider causal relations among cause-and-effect inferences, the decoders for *intent* and *after* are connected to those of *before* and *intent*, respectively. Through causal connections between the three decoders, it can attend to the hidden states of the former decoders, which take the role of *cause* to generate *effect* captions (i.e., the proposed intent/after decoder can attend to not only the hidden states of the transformer encoder, but also the hidden states of the before/intent decoder).

The overall contribution and novelty of this work are summarized as follows: (1) We propose CE-BART, which is a novel transformer-based reasoning pipeline to handle both comprehensive understandings of multimodal input and cause-and-effect caption generation. (2) We extended BART with a graph-based information encoder and to have three decoders in order to address the issues of existing VCG methods. (3) We empirically show that our proposed CE-BART is the state-of-the-art on the VisualCOMET and AVSD benchmarks.

## 2. Related Works

### 2.1. Commonsense Reasoning

Commonsense knowledge has attracted a great deal of attention in both the computer vision and natural language communities. Commonsense or causality knowledge refers to the basic level of practical knowledge and reasoning about everyday situations and events commonly shared among most people [6,8]. For example, if the Sun is out, it is unlikely to rain; if we drop a cup, it is likely to break. Such causality knowledge has been shown to be beneficial for many tasks [9,10], and thus, it is essential for machines to learn to understand causality [11].

In the field of natural language processing, several commonsense Knowledge Bases (KBs) have been constructed to help machines better understand commonsense causality. ConceptNet [12] and ATOMIC [13] are widely used commonsense KBs that leverage human annotations to provide high-quality causality knowledge. These KBs are built based on tuples (s,r,o), where *s* and *o* are subject and object phrases and *r* defines the relation between them. Relations in commonsense KBs include *causes, because, before, as a result, …*, which are essential for learning causality. Bosselut et al. [14] proposed COMET, a transformer-based architecture, for automatic commonsense knowledge base completion. COMET is trained to predict the object *o*, given subject *s* and relation *r*. In the field of computer vision, the Visual Commonsense Reasoning (VCR) task [4] has been proposed, which is a visual question-answering benchmark that requires the machine to provide a rationale explaining why its answer is correct.

### 2.2. Visual Commonsense Generation

Park et al. [1] proposed the task of visual commonsense generation and a corresponding benchmark, VisualCOMET, which aims to generate cause-and-effect descriptions for a given image and corresponding textual event and place. VisualCOMET is a visual commonsense knowledge base where an image and corresponding textual event and place take the place of the object in ATOMIC. There are only a few works [1,6] dealing with the task of visual commonsense generation. Park et al. [1] first proposed a baseline model based on GPT-2 [15]. The baseline model feeds visual and textual context as inputs and is trained to predict each of the cause-and-effect descriptions. Xing et al. [6] proposed Knowledge-enhanced Multimodal BART (KM-BART), which utilizes a BART [16] to pre-train on large external datasets and leverages knowledge from them. KM-BART was first pre-trained with knowledge-based commonsense generation by leveraging knowledge from COMET [14], attribute and relation prediction using the Visual Genome benchmark [17], and masked language and region modeling using various pre-training benchmarks. It was then fine-tuned on the VisualCOMET benchmark to achieve state-of-the-art performance on the VisualCOMET benchmark. However, we argue that these systems operate on the conventional learning scheme of visual and textual information, overlooking the distinctiveness of the cause-and-effect generation task, and possess two major limitations: (1) conventional vision–language transformers are directly utilized to learn relationships between input modalities; (2) every training example is trained independently without considering the relations with others.

## 3. Cause-and-Effect BART

First, we provide a formal definition of the visual commonsense generation task [1] as follows. We are given tuples of (v,e,p), consisting of an image *v*, the event description *e*, and the place description *p*. The goal of visual commonsense generation is to generate three cause-and-effect captions corresponding to (1) what needed to happen *before*, (2) what is the current *intent* of the person, and (3) what will happen *after*.

Figure 2 shows a schematic of Cause-and-Effect BART (CE-BART), consisting of a multimodal input encoder, structured graph reasoner, and cause-and-effect generator. The multimodal input encoder first embeds three input modalities into the feature space. Then, the structured graph reasoner captures intra- and inter-modality relationships among the input modalities. Finally, the cause-and-effect generator generates cause-and-effect captions by considering the causal relationships among inferences. All three components of CE-BART are elaborated on in the following subsections.

### 3.1. Multimodal Input Encoder

Following the previous work on visual commonsense generation, we used the mask R-CNN [18] to detect the visual person, which extracts $N^{v}$ number of appearance features $A={\left\{a_{i}\right\}}_{i=1}^{N^{v}}$ and their corresponding location features $B={\left\{b_{i}\right\}}_{i=1}^{N^{v}}$. Each location feature $b_{i}=\left[x_{i},y_{i},w_{i},h_{i}\right]$ represents a spatial coordinate, where $\left[x_{i},y_{i}\right]$ denotes the relative coordinate of the top-left point of the bounding box, while $\left[w_{i},h_{i}\right]$ denotes the width and height of the box. We calculate the final visual feature as $F^{v}={\left\{v_{i}\right\}}_{i=1}^{N^{v}}\in R^{N^{v}\times d_{v}}$, where $v_{i}=w^{a}a_{i}+w^{b}b_{i}$ and $w^{a},w^{b}$ are learnable weights that embed both features into the visual feature dimension $d_{v}$.

There are two types of text for each image (i.e., event *e* and place *p*). Each sentence for event and place is fed into the word-embedding layer of the pre-trained BART to be further utilized. We obtain the textual feature as follows: $F^{e}={\left\{e_{i}\right\}}_{i=1}^{N^{e}}$, $F^{p}={\left\{p_{i}\right\}}_{i=1}^{N^{p}}$, where $N^{v},N^{p}$ are the number of token features, and $e_{i},p_{i}\in R^{d_{t}}$ are the embedding of the *i*-th token in the event and place, respectively. Further, Table 1 summarizes the descriptions of the important symbols used in this paper.

### 3.2. Structured Graph Reasoner

In order to capture the intra-modality relationships from individual modalities (i.e., image, event, and place) and the inter-modality relationships among the input modalities, the structured graph reasoner first builds semantic graphs for each modality: an image semantic graph $G^{v}$, an event semantic graph $G^{e}$, and a place semantic graph $G^{p}$. Motivated by [19], which projects visual features in the spatial domain into the graph domain for relational reasoning over a global context, the structured graph reasoner performs graph convolutions to capture intra-modality relations. It then captures the higher-order semantic relations among graphs (i.e., inter-modality relationships) via tripartite graph attention to strengthen the multimodal graph representations. The final strengthened semantic representations ($S^{v},S^{e},S^{p}$) are fed into the following cause-and-effect generator.

For simplicity, we denote the feature representation as $F^{x}$ and the semantic graph as $G^{x}$ for each modality x∈{v,e,p}. We first project the feature representation $F^{x}$ into semantic graph $G^{x}$, which is a lightweight *fully connected* graph. Since we will directly reason over graph nodes, the projection into the graph domain is formulated as a linear combination among input features. A linear combination over input features can be thought of as weighted global pooling or global attention. We divide graph projection into two parts: dimension reduction and graph projection. In dimension reduction, we embed all three modality features into the same small feature space. In graph projection, we compute the weights for weighted global pooling. Finally, embedded features are weighted globally pooled to form the graph node feature:(1)$$\begin{array}{cc} G^{x} & =f^{proj}\left(F^{x};W_{f^{proj}}\right)\times f^{reduc}\left(F^{x};W_{f^{reduc}}\right)\in R^{N^{x}\times d_{g}}, \end{array}$$
where the dimension reduction function $f^{reduc}$ parameterized by $W_{f^{reduc}}$ projects each feature into the graph feature dimension $d_{g}$ and the graph projection function $f^{proj}$ parameterized by $W_{f^{proj}}$ produces the weights for linear combination. Here, both $f^{reduc}\;\mathrm{and}\;f^{proj}$ are 1D convolution layers with a kernel size of 1.

In order to capture intra-modality relations in individual semantic graphs, we utilize graph convolution [20] to update node representations and obtain ${\bar{G}}^{x}$. We reason over fully connected graphs by learning the edge weights, which model the interactions among globally pooled graph node features of each modality. Given a fully connected graph $G^{x}$, graph convolution learns the edge weights that correspond to the correlations between node representations. We divide graph convolution into two parts: channelwise convolution and nodewise convolution. By implementing 2D convolution with two 1D convolutions, graph convolution can be more efficient. A single layer of graph convolution is formulated as:(2)$$\begin{array}{c} {\bar{G}}^{x}=\Lambda G^{x}W_{x}=\left(\left(I-A^{x}\right)G^{x}\right)W_{x}, \end{array}$$
where Λ and $A^{x}$ are $N^{x}\times N^{x}$ an adjacency matrix for diffusing information across nodes of $G^{x}$, $W_{x}\in R^{d_{g}\times d_{g}}$ denotes the state update weight, and $I\in R^{N^{x}\times N^{x}}$ is the identity matrix. Here, the adjacency matrix $A^{x}$ is randomly initialized and learned during training, together with $W_{x}$, and the identity matrix serves as a shortcut connection. We can implement Equation (Equation 2) using two consecutive 1D convolution layers along different directions: channelwise convolution (i.e., modeling $\left(I-A^{x}\right)$) and nodewise convolution (i.e., modeling $W_{x}$).

Finally, we capture inter-modality relations among the three semantic graphs (${\bar{G}}^{v},{\bar{G}}^{e},{\bar{G}}^{p}$) via tripartite graph attention and calculate the strengthened semantic representations ($S^{v},S^{e},S^{p}$). We perform graph attention over a *tripartite* graph, which connects all the nodes in individual modalities to all the nodes belonging to the other modalities. By doing so, every node in each modality learns to integrate informative semantics from the other modalities into its representation in order to effectively capture inter-modality relations. First, we concatenate ${\bar{G}}^{v},{\bar{G}}^{e},{\bar{G}}^{p}$ along the node axis to make a tripartite graph structure ${\bar{G}}^{T}$, where each node is connected to all other nodes belonging to different modalities: ${\bar{G}}^{T}=[{\bar{G}}^{v}\left|\right|{\bar{G}}^{e}\left|\right|{\bar{G}}^{p}]\in R^{N^{T}\times d_{g}}$. We perform graph attention [21] over ${\bar{G}}^{T}$ to calculate the multi-head attention to capture relations between each node and its neighboring nodes (i.e., inter-modality relations):(3)$$\begin{array}{cc} S^{T} & =\mathrm{GAT}({\bar{G}}^{T}), \end{array}$$(4)$$\begin{array}{cc} S^{v},S^{e},S^{p} & =slice(S^{T}), \end{array}$$
where the slice(·) operation slices the multimodal representations along the node axis with the corresponding length of each modality. Graph attention effectively models the interaction among graph nodes through the self-attention mechanism inside. Basically, it calculates self-attention over neighboring nodes with the residual connection. Since the proposed tripartite graph attention is performed over the tripartite graph, we can effectively diffuse semantic information to different modalities.

### 3.3. Cause-and-Effect Generator

A Cause-and-Effect Generator (CEG) is proposed to generate cause-and-effect captions by considering the causal relationships among inferences. It is a sequence-to-sequence transformer architecture that feeds the strengthened semantic graph ($S^{v},S^{e},S^{p}$) and decodes cause-and-effect captions (*before*
$O^{b}$, *intent*
$O^{i}$, and *after*
$O^{a}$) in an autoregressive manner. In contrast to existing approaches that treat the generation of each caption as a separate objective, the cause-and-effect generator infers all three cause-and-effect captions holistically. Formally, we have the function of the CEG $f_{CEG}$ with its parameter $W_{f_{CEG}}$, whose goal is:(5)$$\begin{array}{c} O^{b},O^{i},O^{a}=f_{CEG}\left(S^{v},S^{e},S^{p};W_{f_{CEG}}\right). \end{array}$$

#### 3.3.1. Encoder

The encoder of the CEG is based on a multi-layer bidirectional transformer, as in BART and its variant in visual commonsense generation, KM-BART. In contrast to the encoder in KM-BART, whose input sequence starts with one of the special tokens <before>, <intent>, or <after> to indicate to the model which cause-and-effect caption should be generated, the CEG only takes the three sets of semantic graph representations (i.e., $S^{v},S^{e},S^{p}$), as it infers all three captions holistically. To inform the encoder about the start and end of different input modalities, we add three sets of special tokens: <b_img>, <e_img> for image embedding $S^{v}$; <b_ev>, <e_ev> for event embedding $S^{e}$; and <b_pl>, <e_pl> for place embedding $S^{p}$.

#### 3.3.2. Decoder

The decoders of the CEG are based on a multi-layer unidirectional transformer as it works in an autoregressive manner during generation. There are a total of three decoders for the CEG—one each for the generation of *before, intent*, and *after* captions. To inform each decoder about the start of generation, we add three special starting tokens for each decoder: <before>, <intent>, and <after>. Further, we add a special end inference token <e_inf> at the end of the target sequence to indicate the stop of a decoding process. During training, we use teacher-forcing [22] to supervise each decoding step, that is ground truth tokens are used as the decoder input. The decoders of the CEG only take a right-shifted target token sequence as the input.

To consider causal relations among cause-and-effect captions, the decoders for *intent* and *after* are connected to those of *before* and *intent*, respectively. Through causal connections between the three decoders, the CEG can attend to the hidden states of the former decoder, which take the role of *cause* to generate *effect* captions (i.e., the proposed *intent/after* decoder can attend to not only the hidden states of the transformer encoder, but also the hidden states of the *before/intent* decoder), as shown in Figure 2. Formally, we divide the function of the CEG $f_{CEG}$ in Equation (Equation 5) as an encoder $E_{CEG}$ and a set of decoders $D_{CEG}^{x}$, where x∈{b,i,a}. The conventional approaches [1,6] generate the cause-and-effect captions separately without considering causal relations:(6)$$\begin{array}{cc} O^{b} & =D_{con}\left(E_{con}\left(v,e,p\right)\right), \end{array}$$(7)$$\begin{array}{cc} O^{i} & =D_{con}\left(E_{con}\left(v,e,p\right)\right), \end{array}$$(8)$$\begin{array}{cc} O^{a} & =D_{con}\left(E_{con}\left(v,e,p\right)\right), \end{array}$$
where $E_{con}$ and $D_{con}$ represent the functions of the encoder and decoder in existing approaches that feed the image *v*, event *e*, and place *p*. On the other hand, the proposed CEG has sequential connections among decoders to preserve causal relations among cause-and-effect captions: (9)$$\begin{array}{cc} O^{b} & =D_{CEG}^{b}\left(E_{CEG}\left(S^{v},S^{e},S^{p}\right)\right), \end{array}$$(10)$$\begin{array}{cc} O^{i} & =D_{CEG}^{i}\left(E_{CEG}\left(\cdot \right),D_{CEG}^{b}\left(\cdot \right)\right), \end{array}$$(11)$$\begin{array}{cc} O^{a} & =D_{CEG}^{a}\left(E_{CEG}\left(\cdot \right),D_{CEG}^{i}\left(\cdot \right)\right). \end{array}$$

## 4. Experiments

### 4.1. Benchmark Dataset

VisualCOMET [1] is a large-scale benchmark dataset for visual commonsense generation and is the only available dataset of its kind at present. It consists of over 1.4 million textual captions of visual commonsense inferences carefully annotated over a diverse set of 59,000 images paired with 139,000 event descriptions. The visual commonsense inferences are divided into 1174k, 146k, and 145k examples for training, validation, and testing, respectively.

Audio Visual Scene-Aware Dialogue (AVSD) [23] provides video, caption, and dialogue history consisting of a series of textual QA pairs, as well as follow-up questions about the video. The goal is to generate a free-form natural language answer to the question. As both tasks share similar input–output relations, CE-BART can be easily applied to video-grounded dialogue and can transfer causal knowledge learned from a visual commonsense generation for better video understanding.

### 4.2. Metrics

For generative evaluation, we follow the official object metrics for the VisualCOMET and AVSD benchmarks, including BLEU [24], METEOR [25], ROUGE-L [26], and CIDEr [27]. The metrics are formulated to compute the word overlapping between each generated caption and reference caption.

The BLEU score is a basic evaluation method often used in natural language processing. It measures the precision between the generated caption and the reference caption by measuring how much the ordered word pairs overlap through n-grams (1 to 4). Although it has obvious limitations because it lacks consideration of grammatical structures and synonyms, it is still widely used.

The METEOR score computes the weighted F-score, which is the harmonic mean of the precision and recall values based on mapping unigrams, which replaces the simple n-gram precision/recall. The reordering penalty term in METEOR penalizes captions that contain the correct words, but in the wrong order.

ROUGE is based on the recall value and is mostly used for text summarization evaluation. Depending on the sequence used for recall computation, ROUGE can be divided into various types; ROUGE-N, ROUGE-L, ROUGE-W, and ROUGE-S. ROUGE-N is based on the n-gram recall value. For instance, ROUGE-1 calculates the recall based on the matching unigram, and so on. ROUGE-L/W/S are based on the Longest Common Subsequence (LCS), weighted LCS, and skip-bigram co-occurrence statistics, respectively. Instead of only using the recall value, they use the F-score based on the corresponding sequence (e.g., the longest common subsequence between the generated and reference caption for ROUGE-l).

The CIDEr score is based on TF-IDF and is proposed for image captioning evaluation. First, the TF-IDF features are calculated for generated and reference captions based on the n-gram. Then, the CIDEr score is calculated by the cosine similarity between two TF-IDF features.

### 4.3. Experimental Details

We initialized the cause-and-effect generator with a pre-trained BART-based model with 6 transformer layers in both the encoder and decoder, and a hidden size of 768. For tripartite graph attention in the structured graph reasoner, the number of heads in multi-head attention was set to 8. We trained using 4 NVIDIA Quadro RTX 8000 (48 GB of memory) and Adam optimizer with $\beta_{1}=0.9$ and $\beta_{2}=0.999$. The learning rate was initially set to 0.0001, and the model was trained up to 30 epochs with an effective training batch size of 512. During inference, we adopted a beam search, and for each set of inputs, we decoded *before*, *intent*, and *after* captions sequentially.

### 4.4. Experimental Results on VisualCOMET

We compared our proposed Cause-and-Effect BART (CE-BART) with state-of-the-art methods on the VisualCOMET benchmark. Table 2 summarizes the experimental results on the VisualCOMET benchmark on both the validation and test splits, since the current state-of-the-art method, KM-BART [6], only provides the results on the validation split. We also provide the results of the ablation study in Table 2 with several variants of CE-BART in order to measure the effectiveness of the proposed key components of CE-BART. All the reported performances in Table 2 are the average values of five independently trained models with different seeds.

Starting from the direct fine-tuning of the BART-based model in VisualCOMET, which showed slightly lower performance than the previous SOTA method KM-BART, every component of the proposed CE-BART demonstrated improved performance on all three metrics. The results of the ablation study suggest that the limitations of existing approaches that we introduced are valid: (1) conventional vision–language transformers are directly utilized to learn relationships between input modalities, and (2) every training example is trained independently without considering relations with others.

The structured graph reasoner is proposed to capture intra- and inter-modality relations among visual and textual representations. The inclusion of graph reasoning led to a 3.21 point boost in the CIDEr metric compared to the BART-based model. Among the components of the structured graph reasoner, intra-modality reasoning provided a 0.68 point gain in CIDEr, and inter-modality reasoning provided a 1.48 point gain. As our inter-modality reasoning module performs multi-head attention over a tripartite graph, whose neighborhood is defined as the nodes of a heterogeneous modality, each node reinforces its representation with the information from other modalities, and thus, it is able to comprehend inter-modality relations effectively. The design of our structured graph reasoner was effective in capturing intra- and inter-modality relations, which is essential in visual commonsense generation.

The cause-and-effect generator is proposed to generate cause-and-effect descriptions holistically by considering the causal relationships among inferences. It improved the CIDEr score by 2.51 points. Through causal connections between three decoders, the cause-and-effect generator looks at the former decoder, which takes the role of cause, to generate an effect description. The design of our cause-and-effect generator was effective in modeling causal relations among generated descriptions, which is essential in a visual commonsense generation.

The results of our ablation study suggest that the proposed CE-BART can effectively capture intra- and inter-modality relationships interspersed in multimodal input representations and can effectively generate cause-and-effect descriptions holistically by considering causal relations through the cause-and-effect generator with causal connections between decoders. In our comparison, CE-BART surpassed the other state-of-the-art methods on both the validation and test splits of the VisualCOMET benchmark. Compared to KM-BART [6], the state-of-the-art method in the validation split, CE-BART obtained a CIDEr score of 43.58, an improvement of nearly 4 points. The CIDEr score of CE-BART (42.64) was nearly double that obtained with the baseline [1], the state-of-the-art method in the test split (17.36). CE-BART also improved the BLEU-2 score by almost 16 points and the METEOR score by more than 7 points.

We also provide an in-depth quantitative analysis of the CEG in Table 3 to show the significance of the dependencies among the three decoders. The motivation behind capturing relations between training samples is to consider the relations among the *before, intent*, and *after* descriptions while generating them for an image: the current *intent* is related to the situation *before*, and the situation *after* is related to the current *intent*. As the CEG has connections among decoders, the *intent* decoder can operate by using not only the image, but also the information from the *before* decoder. Similarly, the *after* decoder can generate predictions with the help of the *intent* decoder. We conducted separate evaluations for *before*/*intent*/*after* captions. Without connections among decoders (i.e., without the CEG), *before* prediction showed superior performance to *intent* and *after* predictions. However, with connections among decoders (i.e., with the CEG), we observed a performance improvement in *intent* and *after* predictions, as expected, obtained by considering relations.

### 4.5. Experimental Results on AVSD

We conducted additional experiments to validate the generalizability of the proposed CE-BART in other VL tasks. We conducted experiments in the task of video-grounded dialogue, which is a multi-turn question-answering task. The formal definition of video-grounded dialogue is as follows: we are given a video, a dialogue history consisting of a series of textual QA pairs, and a follow-up question about the video, and the goal is to generate a free-form natural language answer. Both tasks share similar input–output relations; therefore, CE-BART can be easily applied to the new task of video-grounded dialogue and can transfer causal knowledge learned from visual commonsense generation for better video understanding. We trained CE-BART for video-grounded dialogue in two settings: (1) CE-BART without pre-training on VisualCOMET and (2) CE-BART with pre-training on VisualCOMET. Through this comparative experiment, we observed that causal information learned from VisualCOMET could help with understanding the video. Table 4 summarizes the results of the AVSD benchmark. We observed that CE-BART showed an improvement over existing methods and achieved SOTA performance in all metrics. By pre-training CE-BART on VisualCOMET, we could further boost the performance, indicating that causal knowledge trained on VisualCOMET can be successfully transferred for better video understanding in AVSD. We were able to validate that our proposed CE-BART can benefit other VL tasks by effectively transferring causal knowledge learned from VisualCOMET.

### 4.6. Qualitative Analysis

Figure 3 shows examples from the test split of VisualCOMET and compares predictions from CE-BART and the baseline. CE-BART successfully utilized the SGR, which captures intra- and inter-modality relationships and is flexible in selecting important information regardless of whether it is textual or visual. In the first example, CE-BART noticed that the text of the father holding his daughter’s shoulder was not crucial information and focused more on the image. With this ability to consider and select crucial information, in the second example, CE-BART focused on the key object in the scene and people’s interaction centered on that object and avoided stating the obvious. Further, as the CEG was trained to successively generate captions through causal connections, it had the information from the former decoder, which took the role of *cause* to generate the *effect* caption. In the third example, CE-BART was continuous regarding the contents, whereas the baseline model produced a discontinuous caption. In the lower-right example, the baseline model produces overlapping captions, while CE-BART effectively generated the rich dynamic story of the visual scene.

## 5. Limitations

We believe that our proposed CE-BART has several limitations that can be addressed through further experiments in the future. First, its scalability is limited due to the requirement of large GPU resources. We conducted experiments using four NVIDIA Quadro RTX 8000 units (48 GB of memory), which are extremely expensive. Second, its scalability to control the time scale is limited. There is no factor in the current task setting that selects how much of a past/future situation is required. We will further develop our methods to overcome the limitations of the model.

## 6. Conclusions

We proposed a novel cause-and-effect BART for the task of visual commonsense generation. The proposed CE-BART consists of two major components: (1) a structured graph reasoner and (2) a cause-and-effect generator. The structured graph reasoner builds semantic graphs for individual modalities and strengthens their representations by capturing intra- and inter-modality relations among graph structures. The cause-and-effect generator is a transformer architecture with three decoders, one each for generating before, intent, and after captions. In experiments on the VisualCOMET and AVSD benchmarks, CE-BART achieved new state-of-the-art performance.

## Figures and Tables

**Figure 1 sensors-22-09399-f001:**
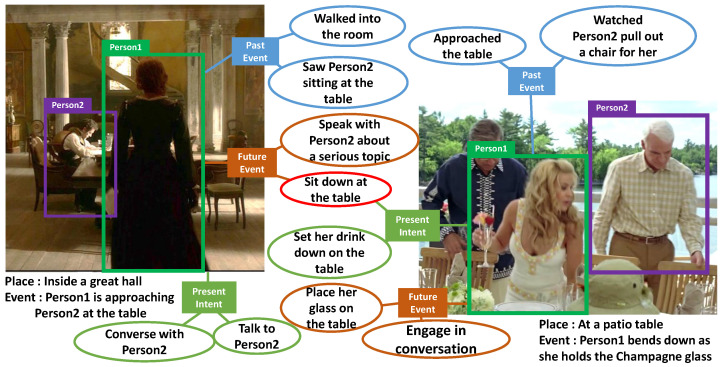
Illustration of visual commonsense generation. Given a person in an image and a corresponding textual event, an agent is required to generate (1) what needed to happen *before*, (2) what is the current *intent* of person, and (3) what will happen *after*.

**Figure 2 sensors-22-09399-f002:**
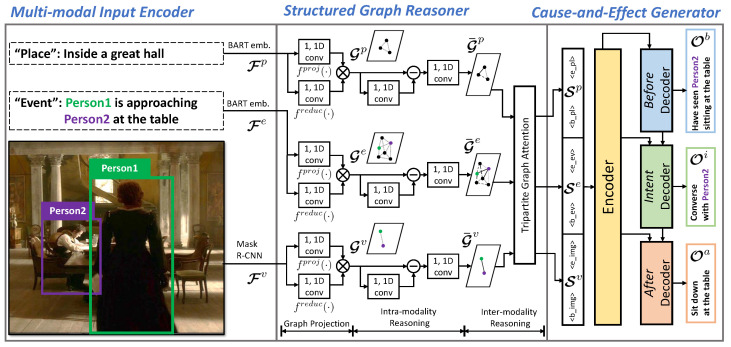
Illustration of Cause-and-Effect BART (CE-BART), which is composed of a multimodal input encoder, structured graph reasoner, and cause-and-effect generator. (1) Multimodal input encoder: we first obtain multimodal features ($F^{v},F^{e},F^{p}$) using pre-trained models. We used mask R-CNN for the visual inputand the BART word-embedding layer for textual inputs. (2) Structured Graph Reasoner (SGR): we then built semantic graphs ($G^{v},G^{e},G^{p}$) and strengthened their representations ($S^{v},S^{e},S^{p}$) by capturing the intra- and inter-modality relations. (3) Cause-and-Effect Generator (CEG): we finally generated cause-and-effect descriptions ($O^{b},O^{i},O^{a}$) using the BART-based transformer architecture, which considers the causal relations among inferences.

**Figure 3 sensors-22-09399-f003:**
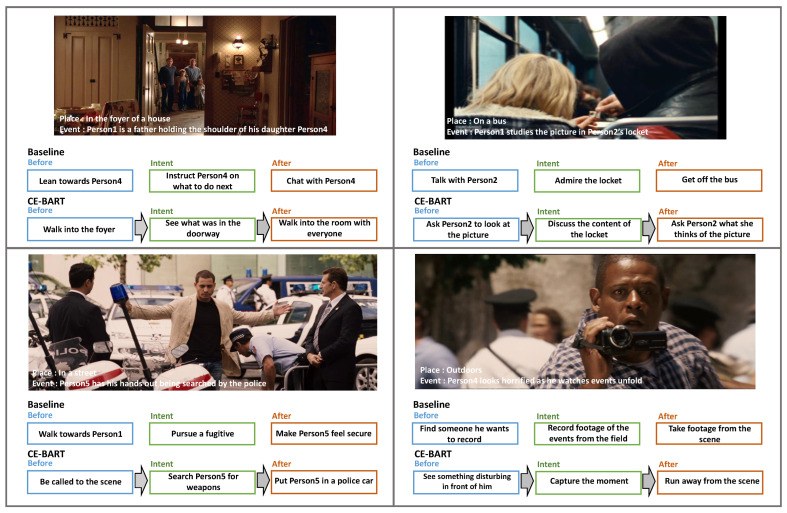
Four examples from the test split of the VisualCOMET benchmark.

**Table 1 sensors-22-09399-t001:** Descriptions of the important symbols used in this paper.

Symbol	Description	Symbol	Description
*v*	Input image	*e*	Input event description
*p*	Input place description	A	Appearance feature
B	Location feature	$F^{v}$	Input visual feature
$F^{e}$	Input event description feature	$F^{p}$	Input place description feature
$G^{v}$	Image semantic graph	$G^{e}$	Event semantic graph
$G^{p}$	Place semantic graph	$S^{v}$	Strengthened semantic image feature
$S^{e}$	Strengthened semantic event feature	$S^{p}$	Strengthened semantic place feature
$O^{b}$	Before caption	$O^{i}$	Intent caption
$O^{a}$	After caption		

**Table 2 sensors-22-09399-t002:** Comparison with state-of-the-art methods on the VisualCOMET benchmark. Here, “Proj-SGR” denotes the graph projection (i.e., Equation (Equation 1)), “Intra-SGR” denotes the intra-modality reasoning (i.e., Equation (Equation 2)), and “Inter-SGR” stands for the inter-modality reasoning (i.e., Equation (Equation 3)). “CEG” stands for the Cause-and-Effect Generator with three decoders (i.e., Equations (Equation 9)–(Equation 11)). The best results are in bold.

Methods	Validation Set			Test Set		
	BLEU2	METEOR	CIDEr	BLEU2	METEOR	CIDEr
Baseline [1]	13.50	11.55	18.27	12.71	11.13	17.36
KM-BART [6]	23.47	15.02	39.76	-	-	-
Variants on CE-BART						
BART-base	22.51	14.73	37.86	-	-	-
+ Proj-SGR	22.47	14.97	38.91	-	-	-
+ Intra-SGR	23.85	15.72	39.59	-	-	-
+ Inter-SGR	25.07	18.24	41.07	-	-	-
+ CEG	28.60	19.32	43.58	-	-	-
CE-BART	**28.60**	**19.32**	**43.58**	**28.14**	**18.91**	**42.64**

**Table 3 sensors-22-09399-t003:** Analysis conducted on the validation split of VisualCOMET. We provide an analysis of the behavior of the CEG by observing separate performance evaluations with (w CEG) and without the CEG (w/o CEG).

Methods	Before			Intent			After		
	B2	M	C	B2	M	C	B2	M	C
w/o CEG	29.7	20.4	45.1	19.4	15.4	40.7	26.1	18.9	37.7
w CEG	30.9	20.9	45.9	25.5	16.6	42.4	29.6	20.2	41.9

**Table 4 sensors-22-09399-t004:** Comparison with state-of-the-art methods on the AVSD benchmark. We compared CE-BART with various state-of-the-art systems on the AVSD benchmark: baseline [28], STSGR [29], MTN [30], MTN-TMT [31], and VX2TEXT [32]. The best results are in bold.

Methods	BLEU1	BLEU2	BLEU3	BLEU4	METEOR	ROUGE-L	CIDEr
Baseline [28]	-	-	-	0.078	0.113	0.277	0.727
STSGR [29]	-	-	-	0.133	0.165	0.362	1.272
MTN [30]	0.356	0.242	0.174	0.135	0.165	0.365	1.366
MTN-TMT [31]	-	-	-	0.142	0.171	0.371	1.357
VX2TEXT [32]	0.361	0.260	0.197	0.154	0.178	0.393	1.605
CE-BART	0.364	0.266	0.203	0.158	0.181	0.400	1.681
CE-BART	**0.365**	**0.268**	**0.205**	**0.161**	**0.183**	**0.404**	**1.721**
w pre-train							

## Data Availability

Not applicable.

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
