# Peer review of "CE-BART: Cause-and-Effect BART for Visual Commonsense Generation"

_sensors, 2022, doi:10.3390/s22239399_

Round 1

Reviewer 1 Report

In this paper, a Cause-and-Effect BART for the task of visual commonsense generation is presented which consists of two major components: (1) Structured Graph Reasoner, and (2) Cause-and-Effect Generator. The results are discussed on VisualCOMET and AVSD benchmarks. CE-BART achieves SOTA performances on both benchmarks,  while extensive ablation study and qualitative analysis demonstrate the performance gain and improved interpretability. The paper is well written and results are discussed in detail. However, there are few comments to further improve the paper:

1)- At the end of Introduction, clearly mention the contribution and novelty of the paper

2)- Add more explanation of the equations to make them more understandable

3)-  Add more explanation of Figure 2.

Author Response

We greatly appreciate the reviewer's detailed and constructive comments. We tried dearly to answer all questions with sincerity.

Point 1: At the end of the Introduction, clearly mention the contribution and novelty of the paper.

Response 1: We summarized and added the contribution and novelty of the paper at the end of the Introduction. Please see lines 85-90 of our revised manuscript.

To be more specific, the following paragraph is added:

The overall contribution and novelty of this work are summarized as follows: (1) We propose CE-BART which is a novel transformer-based reasoning pipeline to handle both comprehensive understandings of multi-modal input and cause-and-effect caption generation, (2) We extended BART with graph-based information encoder and to have three decoders in order to address issues of existing VCG methods, (3) We empirically show that our proposed CE-BART is state-of-the-art in VisualCOMET and AVSD benchmarks.

Point 2: Add more explanation of the equations to make them more understandable.

Response 2: We added more explanations to the equations to make them more understandable. Please see lines 173-180 of our revised manuscript for an explanation of Equation 1. Please see lines 186-192 of our revised manuscript for an explanation of Equation 2. Please see lines 213-217 of our revised manuscript for an explanation of Equation 3.

To be more specific, the following explanations are added:

Equation 1: Since we will directly reason over graph nodes, the projection into the graph domain is formulated as a linear combination among input features. Linear combination over input features can be thought of as weighted global pooling or global attention. We divide graph projection into two parts; dimension reduction and graph projection. In dimension reduction, we embed all three modality features into the same small feature space. In graph projection, we compute the weights for weighted global pooling. Finally, embedded features are weighted globally pooled to form graph node feature:

Equation 2: We reason over fully-connected graphs by learning edge weights which model the interactions among globally pooled graph node features of each modality. We divide graph convolution into two parts; channel-wise convolution and node-wise convolution. By implementing 2D convolution with two 1D convolutions, graph convolution can be more efficient.

Equation 3: Graph attention effectively models the interaction among graph nodes through the self-attention mechanism inside. Basically, it calculates self-attention over neighboring nodes with the residual connection. Since the proposed tripartite graph attention is performed over the tripartite graph, we can effectively diffuse semantic information to different modalities.

Point 3: Add more explanation of Figure 2.

Response 3: We added the explanation of Figure 2. Please see lines 139-145 of our revised manuscript.

To be more specific, the following paragraph is added:

Figure 2 shows a schematic of Cause-and-Effect BART (CE-BART), consisting of a multi-modal input encoder, structured graph reasoner, and cause-and-effect generator. Multi-modal input encoder first embeds three input modalities into feature space. Then, Structured graph reasoner captures intra- and inter-modality relationships among input modalities. Finally, Cause-and-effect generator generates cause-and-effect captions by considering the causal relationships among inferences. All three components of CE-BART are elaborated on in the following subsections.

Reviewer 2 Report

1. The paper is too short to justify the experiment on VisualCOMET and AVSD dataset

2. The explanation of the proposed work could be improved. 

3. Please add a table explaining all the symbols and notations used in the work

4. Details flow for Multimodal input encoder, Structure graph reasoner and cause & effect generator are required to benefit the readers. 

5. Instead of referring to Graph projection, Intar & Inter modality reasoning, please include some explanation in the paper. 

6. Please add descriptions for all the metrics used for evaluation 

7. How is the weakness of Bleu score overcome (or) The weakness of Bleu doesn't impact the proposed  work

8. Line 157 how fredc , fproj kernel size was determined 

9. Please use the standard reference style to refer figures ( Fig or Figure )

10. Please note: would suggest to Add the real experimental results of work instead of Figure 3.

Author Response

We greatly appreciate the reviewer's detailed and constructive comments. We tried dearly to answer all questions with sincerity.

Points 1 & 2 & 5: The paper is too short to justify the experiment on VisualCOMET and AVSD dataset. The explanation of the proposed work could be improved. Instead of referring to Graph projection, Intra- & Inter-modality reasoning, please include some explanation in the paper. 

Responses 1 & 2 & 5: We added more explanations to Section 3 to make them more understandable. Please see lines 173-180 of our revised manuscript for an explanation of Equation 1. Please see lines 186-192 of our revised manuscript for an explanation of Equation 2. Please see lines 213-217 of our revised manuscript for an explanation of Equation 3.

To be more specific, the following explanations are added:

Equation 1: Since we will directly reason over graph nodes, the projection into the graph domain is formulated as a linear combination among input features. Linear combination over input features can be thought of as weighted global pooling or global attention. We divide graph projection into two parts; dimension reduction and graph projection. In dimension reduction, we embed all three modality features into the same small feature space. In graph projection, we compute the weights for weighted global pooling. Finally, embedded features are weighted globally pooled to form graph node feature:

Equation 2: We reason over fully-connected graphs by learning edge weights which model the interactions among globally pooled graph node features of each modality. We divide graph convolution into two parts; channel-wise convolution and node-wise convolution. By implementing 2D convolution with two 1D convolutions, graph convolution can be more efficient.

Equation 3: Graph attention effectively models the interaction among graph nodes through the self-attention mechanism inside. Basically, it calculates self-attention over neighboring nodes with the residual connection. Since the proposed tripartite graph attention is performed over the tripartite graph, we can effectively diffuse semantic information to different modalities.

Point 3: Please add a table explaining all the symbols and notations used in the work.

Response 3: We added a table describing important symbols used in the paper. Please see Table 1 of our revised manuscript.

Point 4: Details flow for Multimodal input encoder, Structure graph reasoner, and cause & effect generator are required to benefit the readers.

Response 4: We added a detailed flow of each modality at the beginning of Section 3. Please see lines 139-145 of our revised manuscript.

To be more specific, the following paragraph is added:

Figure 2 shows a schematic of Cause-and-Effect BART (CE-BART), consisting of a multi-modal input encoder, structured graph reasoner, and cause-and-effect generator. Multi-modal input encoder first embeds three input modalities into feature space. Then, Structured graph reasoner captures intra- and inter-modality relationships among input modalities. Finally, Cause-and-effect generator generates cause-and-effect captions by considering the causal relationships among inferences. All three components of CE-BART are elaborated on in the following subsections.

Point 6: Please add descriptions for all the metrics used for evaluation 

Response 6: We added descriptions for all the metrics used for evaluation in subsection 4.2 of our revised manuscript. Please lines 274-298 of our revised manuscript.

To be more specific, the following subsection is added:

4.2. Metrics

For generative evaluation, we follow the official object metrics for VisualCOMET and AVSD benchmarks, including BLEU [27], METEOR [28], ROUGE-L [29], and CIDEr [30]. The metrics are formulated to compute the word overlapping between each generated caption and reference caption.

The BLEU score is a basic evaluation method often used in natural language processing. It measures the precision between the generated caption and the reference caption by measuring how much the ordered word pairs overlap through n-grams (1 to 4). Although it has obvious limitations because it lacks consideration for grammatical structures and synonyms, it is still widely used.

The METEOR score computes weighted F-score which is the harmonic mean of precision and recall values based on mapping unigrams which replaces the simple n-gram precision/recall. Reordering penalty term in METEOR penalizes captions that contain the correct words, but in wrong order.

The ROUGE is based on recall value, and is mostly used for text summarization evaluation. Depending on the sequence used for recall computation, ROUGE can be divided into various tyopes; ROUGE-N, ROUGE-L, ROUGE-W, and ROUGE-S. The ROUGE-N is based on n-gram recall value. For instance, ROUGE-1 calculates recall based on matching unigram, and so on. The ROUGE-L/W/S are based on longest common subsequence (LCS), weighted LCS, and skip-bigram co-occurrence statistics, respectively. Instead of only using recall value, they use F-score based on corresponding sequence. (e.g., longest common subsequence between generated and reference caption for ROUGE-l)

The CIDEr score is based on TF-IDF and is proposed for image captioning evaluation. First, TF-IDF features are calculated for generated and reference captions based on n-gram. Then, CIDEr score is calculated by cosine similarity between two TF-IDF features.

Point 7: How is the weakness of Bleu score overcome (or) The weakness of Bleu doesn't impact the proposed  work

Response 7: BLEU score is a very basic evaluation metric used in various fields of NLP. Every previous approach used the BLEU score to measure performance. Thus, we also used the BLEU score to measure the performance of CE-BART and compare it with previous approaches. As a result, the proposed CE-BART surpass previous approaches in every metrics including BLEU score.

Point 8: Line 157 how f^{reduc} , f^{proj} kernel size was determined 

Response 8: It is determined empirically. We experimented with several variants of CE-BART with different sizes of kernels. There was no difference in performance regarding the size of kernels. Thus, we selected a kernel size of 1 to make our model computationally efficient.

Point 9: Please use the standard reference style to refer to figures ( Fig or Figure )

Response 9: We utilized the standard reference style to refer to figures as Figure X. Please see line 23 of our revised manuscript.

Point 10: Please note: would suggest adding the real experimental results of work instead of Figure 3.

Response 10: Figure 3 is composed of real experimental results on the VisualCOMET benchmark. We randomly selected data samples from the test split of VisualCOMET and generated three cause-and-effect captions. These captions are used to make Figure 3.

Round 2

Reviewer 2 Report

Ok with the authors response on the review comments.